# Serial choriocapillaris flow changes in eyes with branched retinal vascular obstruction (BRVO)

**Bo-Een Hwang**[1,2]**, Jae-Hyuck Kwak**[1,2]**, Joo-Young Kim**[1,2]**, Rae-Young Kim**[1,2]**, Mirinae Kim**[1,2]**, Young-Geun Park**[1,2]**, Young-Hoon Park** [1,2]*****

1 Department of Ophthalmology and Visual Science, Seoul St. Mary's Hospital, College of Medicine, The Catholic University of Korea, Seoul, Korea, 2 Catholic Institute for Visual Science, College of Medicine, The Catholic University of Korea, Seoul, Korea

* parkyh@catholic.ac.kr

## Abstract

### Purpose

To evaluate the choriocapillaris (CC) flow changes in branched retinal vascular obstruction (BRVO) on optical coherence tomography angiography (OCTA).

### Methods

Retrospective analysis of 29 patients with unilateral BRVO (58 eyes, including unaffected fellow eyes for controls). 4.5 x 4.5 mm macular scans were divided into 4 quadrants. Serial analyses were conducted on CC flow voids of the opposite quadrant to the active occluded area in BRVO eyes. Each of the quadrants were also compared to the occluded quadrant of resolved eyes and the contralateral quadrant of fellow eyes by matched data analysis. A regression analysis was performed on the several parameters (Choroidal thickness; CT, Choroidal vascularity index; CVI, Central macular thickness; CMT, The number of intravitreal injections) and CC flow voids.

### Results

The CC flow void increased sequentially: The uninvolved quadrant of acute BRVO-affected eyes, that of resolved eyes after 3-month/1-year, the contralateral quadrant of fellow eye, the involved (occluded) quadrant of resolved eyes. There were significant correlations between initial CMT, the number of injections and the CC flow void of uninvolved quadrants (P = 0.025, 0.031, respectively), and between the involved (occluded) quadrants and fellow CT (P = 0.029).

### Conclusion

CC flow void of uninvolved macular areas decreased significantly in eyes with acute BRVO, suggesting that CC changes were limited to the blocked area and a compensatory mechanism would work in surrounding areas.

**Funding:** This study was supported by the Basic Science Research Program through the National Research Foundation of Korea (NRF-2020R1F1A1074898). The funders had no role in study design, data collection and analysis, decision to publish, or preparation of the manuscript.

**Competing interests:** The authors have declared that no competing interests exist.

## Introduction

Branched retinal vein occlusion (BRVO) is a vision-threatening disease depending on ischemic area and recurrence of macular edema (ME) [1, 2]. Although BRVO is a disease of the retina, several studies presented some remarkable results that choroidal characteristics would affect the disease prognosis [3–6]. Prognosis of BRVO is mainly determined by visual acuity, which is also affected by outer retina integrity and choriocapillaris (CC) flow. CC flow was decreased in chronic BRVO in some research [7–9], but it is not clear whether this change is due to the inflow of extracellular fluid or that of vascular endothelial growth factor (VEGF) from retina to choroid [7, 10].

Due to the development of OCTA, recent studies on CC have been vigorously conducted, but limited research has been done on BRVO. Even if there were, the analyses have been done only for the chronic and stable BRVO without ME. The reason was that in the case of acute BRVO, active hemorrhage and severe edema often involved macula, and accurate CC flow analysis was challenging.

However, in most acute BRVOs, the degree of hemorrhage and ME is particularly severe in any quadrant in OCTA macula scan, rather than the entire macula, and in the opposite quadrant, it is often not severe, even close to normal. Therefore, this study identified how CC flow voids in uninvolved quadrants changed from acute affected BRVO eyes to short-term resolved eyes and long-term resolved eyes using Topcon SS-OCTA (DRI Triton) macula scans. It is thought that it would be possible to obtain a lot of information about changes in choroidal flow in BRVO. We also conducted some analyses to ascertain whether there was any association between CC flow voids and optical coherence tomography (OCT) quantitative parameters, such as choroidal thickness (CT) and choroidal vascularity index (CVI), and the central macular thickness (CMT), to better understand the pathophysiology of BRVO.

## Methods

### Study population

The observational retrospective study was performed in the Department of Ophthalmology and Visual Science at Seoul St Mary's Hospital and adhered to the tenets of the Declaration of Helsinki. All protocols were approved by the Institutional Review Board (IRB) of Seoul St. Mary's Hospital, The Catholic University of Korea Catholic Medical Center. (KC22RISI0102) Due to the retrospective nature and anonymized data, the written consent procedure was exempted in accordance with the Seoul St. Mary's Hospital IRB regulations.

Twenty-nine patients who were diagnosed with monocular BRVO were selected in this study; their corresponding healthy fellow eyes were also analyzed. All participants were chosen between January 2019 and August 2020 at Seoul St. Mary's Hospital in Korea. We performed a retrospective review of their medical records. The exclusion criteria were: (1) refractive errors (spherical equivalent, ± six diopters); (2) history of ocular trauma or treatment (laser treatment, intraocular surgery, intravitreal injections); (3) systemic diseases that could make a retinal change except hypertension; (4) other retinal illness, including age-related macular degeneration, diabetic retinopathy, uveitis, glaucoma, or other neurodegenerative disease; (5) media opacities that could alter image quality.

### Study protocol

Demographic data, medical history, and ophthalmologic history were recorded at the initial visit. All subjects underwent an ocular examination, including slit-lamp microscopy, dilated fundus examination, OCT, and OCTA. OCT/OCTA imaging was performed using the

Topcon DRI Triton SS-OCT device using a 1050-nm wavelength light source, and a scanning speed of 100,000 A-scans/s. BRVO was diagnosed when its typical characteristics (regional flame shift hemorrhage along with vessels, macula edema) were present on fundus exam and OCT images. The BRVO eyes accompanying vitreous hemorrhage or multiple vascular obstruction (ex. hemi-CRVO) were excluded. 4.5 x 4.5 mm OCTA CC macular scans were divided into 4 quadrants. Serial analyses were conducted on CC flow voids of the opposite (uninvolved) quadrant to the active vessel-occluded area in BRVO eyes. OCT/OCTA images showing retinal hemorrhage with ME, involving macula no more than half, were selected and designated as "Acute BRVO-affected eye". Resolutions of ME and macular hemorrhage were defined to reduced central retinal thickness by more than a half or decreased retinal hemorrhage with intraretinal fluid by more than two-thirds within 3 months after any intravitreal injections. The resolved states were confirmed in all subjects and OCT/OCTA images at that time were used for the analysis of "Short-term BRVO-resolved eyes". "Long-term BRVO-resolved eyes" pertained to the images of stable BRVO without ME and hemorrhage after 1-year disease onset. "Fellow eyes" pertained to the contralateral quadrant CC images from the other unaffected fellow eyes of BRVO patients. "Long-term occluded (involved) BRVO-resolved eyes" pertained to the images of the vessel-occluded (involved) quadrants of resolved stable BRVO after 1-year disease onset. All OCTA images were evaluated by two experienced independent retinal specialists (Y-H.P. and B-E.H.) who were blinded to the other imaging findings and the clinical histories.

## Choriocapillaris flow void measurement

All OCTA images in this study met an image quality of 65 or higher with no line artifacts or noise. Fig 1A–1C shows the process of calculation of a CC void image in OCTA. The 4.5 × 4.5 mm (320 × 320 pixels) CC images were obtained using a slab from the boundary of the basement membrane (BM) to 20.8 μm, which was calculated from the OCTARA segmentation algorithm built in the Topcon imageNET software.

We selected a quadrant square area of 2.25 × 2.25 mm (160 × 160 pixels) as the opposite (uninvolved) lesion, which is located on the opposite side of the area that showed black pixels

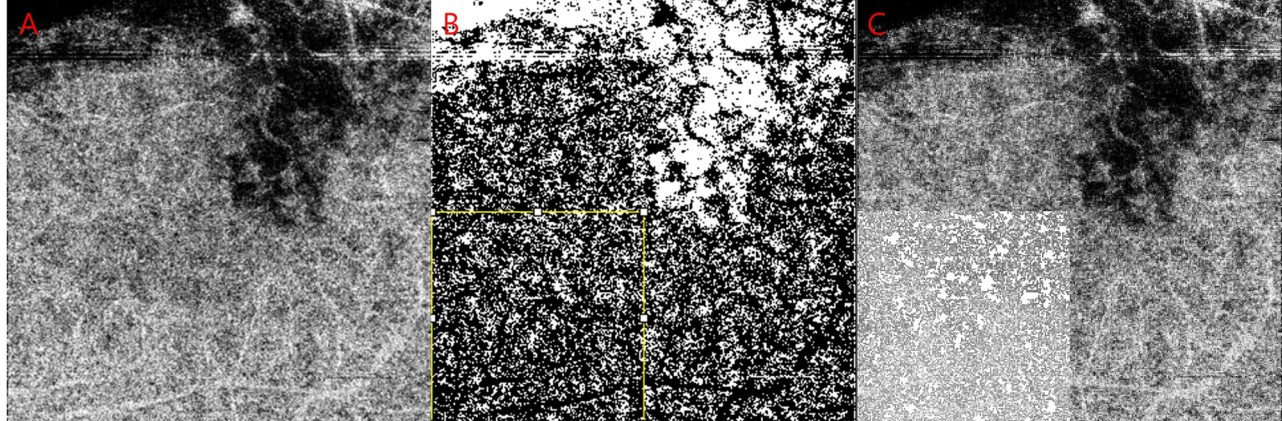

**Fig 1. A representative case of affected eyes to demonstrate the process to calculate choriocapillaris (CC) flow voids (%) of the opposite uninvolved quadrant in the OCTA image.** (A) The 4.5 × 4.5 mm (320 × 320 pixels) CC images were obtained using a slab from the boundary of the basement membrane (BM) to 20.8 μm. (B) A quadrant square area of 2.25 × 2.25 mm (160 × 160 pixels) was selected as the opposite uninvolved lesion (yellow), which is located on the opposite side of the area that showed black pixels due to bleeding from vessel obstructions (involved). Then, we applied the phansalkar threshold method (15 window radius) to the uninvolved quadrant. (C) Calculated flow voids using Image J tool "Analyze particle" (>1 pixel) were presented as white only within the boundary of the quadrant.

due to bleeding from vessel obstructions (involved). In long-term (1-year disease onset) BRVO-resolved eyes, the same-sized occluded quadrants were also selected. Then, we applied the phansalkar threshold method (15 window radius) to the uninvolved and involved quadrants, respectively [11, 12]. Using the "Analyze particle" command in Image J software (version 1.53a; https://imagej.nih.gov/ij/) that calculated all threshold areas bigger than or equal to 1 pixel of flow void, the percentage (%), count (n), average size (pixel^2) of CC flow void area were measured.

## Choroidal thickness measurement

CT was calculated using the automatic built-in software within the SS-OCT device. Subfoveal CT was determined by calculating the distance from the outer border of the RPE to the inner edge of the suprachoroidal space [13]. We measured CT manually at the foveal center using digital calipers provided by the SS-OCT software. Two experienced independent observers measured the SFCT, and the average value was used in the analysis to avoid inter-observer variation.

## Choroidal vascularity index assessment

A 12 mm raster scan passing through the fovea was chosen for image binarization to obtain the CVI. It was segmented using the protocol described by Agrawal et al. [14], and image binarization was performed using Image J software. The ratio of luminal area (LA) to total choroidal area (TCA) was defined as the CVI. Two experienced independent observers measured the CVI, and the average value was used in the analysis to avoid inter-observer variation.

## Statistical analysis

Statistical analysis was performed using the Statistical Package for the Social Sciences for Windows (version 24.0; SPSS, Inc., Chicago, IL). The intraclass correlation coefficient (ICC) analysis was conducted to evaluate intergrader reliability. The mean differences between uninvolved quadrants of the acute BRVO-affected eye, short-term BRVO-resolved eyes, long-term BRVO-resolved eyes, occluded (involved) quadrants of long-term BRVO-resolved eyes, and the fellow eyes were assessed using the paired t-test. Modulated p-value was calculated by multiplying the number of comparisons (x10) to the original p-value from the paired t-test. A linear regression analysis was conducted between the OCT parameters (CVI, CT, CMT), the number of intravitreal injections and the quadrant CC flow voids (%). Two-sided $P$-values of <0.05 were considered to be statistically significant.

## Results

Demographics and characteristics, including OCT and OCTA measurements (CMT, CVI, CT and CC) are in Table 1. ICC was 0.972 for single measures and 0.986 for average measurements (p < 0.001), which were assessed as an excellent agreement between two graders [15]. The mean quadrant CC flow void increased significantly, with the following sequence: the uninvolved quadrant of acute BRVO-affected eyes, that of resolved eyes after 3-months, 1-year disease onset, the contralateral quadrant of fellow eye, the involved quadrant of resolved eyes after 1-year disease onset. Comparisons between each quadrant were statistically significant ($P$<0.01), with the exception of some resolved eyes' comparisons ($P$>0.05). (Table 2, Fig 2).

The results of the linear regression analysis between the OCT choroidal parameters (CMT, CT, fellow CVI), total intravitreal injections (anti-VEGFs and steroids) and the OCTA CC flow void are shown in Table 3. There were significant correlations between initial CMT, the

**Table 1. Demographics and characteristics of the study subjects.**

| Demographics and characteristics | BRVO patients (n = 29) |
|---|---|
| Age, years | 61.34 (±12.79) |
| Sex, male:female | 12:17 |
| Disease eye, OD:OS | 16:13 |
| Hypertension (%) | 34 |
| Intravitreal injections (n) | 2.76 (±1.84) |
| **Choriocapillaris void in OCTA** | |
| Affected uninvolved quadrant CC void (%) | 27.03 (±6.46) |
| Resolved uninvolved quadrant CC void (%) (3-months) | 35.85 (±4.09) |
| Resolved uninvolved quadrant CC void (%) (1-year) | 35.84 (±5.85) |
| Resolved occluded quadrant CC void (%) (1-year) | 43.10(±4.54) |
| Fellow control quadrant CC void (%) | 38.99 (±1.80) |
| **OCT parameter** | |
| Affected eye initial CMT (μm) | 554.76 (±208.29) |
| Affected eye initial CT (μm) | 314.07 (±78.34) |
| Fellow eye CT (μm) | 308.34 (±73.96) |
| Fellow eye CVI (%) | 64.63 (±2.10) |

CC: choriocapillaris; CMT: central macular thickness; CVI: choroidal vascularity index; CT: subfoveal choroidal thickness. Data are presented as the mean (± SD) or a number, as appropriate.

number of intravitreal injections and CC flow void of acute BRVO uninvolved quadrants (P = 0.025, 0.031, respectively), and between the involved (occluded) quadrants of resolved eyes after 1-year disease onset and fellow CT (P = 0.029). Otherwise, no statistically significant relationship was found.

## Discussion

The significant decrease in CC flow void in the uninvolved quadrant of acute BRVO affected eyes indicates that there might be a compensation of surrounding areas for the CC flow reduction in the obstructed region by using VEGF effects or autonomic nerve systems [16, 17].

**Table 2. Choriocapillaris flow void correlation results.**

| | Affected eye | Resolved eye | Resolved eye | Resolved eye | Fellow eye |
|---|---|---|---|---|---|
| | (U/Q) | (U/Q) | (U/Q) | (O/Q) | (C/Q) |
| | (baseline) | (3 months) | (1 year) | (1 year) | (baseline) |
| | 1 | 2 | 3 | 4 | 5 |
| CC void (%) | 27.03 (±6.46) | 35.85 (±4.09) | 35.84 (±5.85) | 43.10(±4.54) | 38.99 (±1.80) |
| Mean difference | 1 vs 2 | 1 vs 3 | 1 vs 4 | 1 vs 5 | 2 vs 3 |
| (Corrected P-value) | **P<0.01** | **P<0.01** | **P<0.01** | **P<0.01** | P = 1.00 |
| | 2 vs 4 | 2 vs 5 | 3 vs 4 | 3 vs 5 | 4 vs 5 |
| | **P<0.01** | P = 0.07 | **P<0.01** | P = 0.06 | **P<0.01** |

Paired t-test for choriocapillaris flow void (%) between the opposite quadrant of BRVO affected eyes, resolved eyes (3 months), long-term resolved eyes (1-year), the occluded quadrant of long-term resolved eyes (1-year) and contralateral quadrant of unaffected fellow eyes.

Bonferonni Correction for multiple comparison testing. Corrected P-value was presented by multiplying the number of comparisons (x10) to the original P-value.

p-values that are statistically significant are highlighted in bold. (p<0.05)

U/Q; uninvolved quadrant, O/Q; occluded quadrant, C/Q; contralateral eye corresponding quadrant.

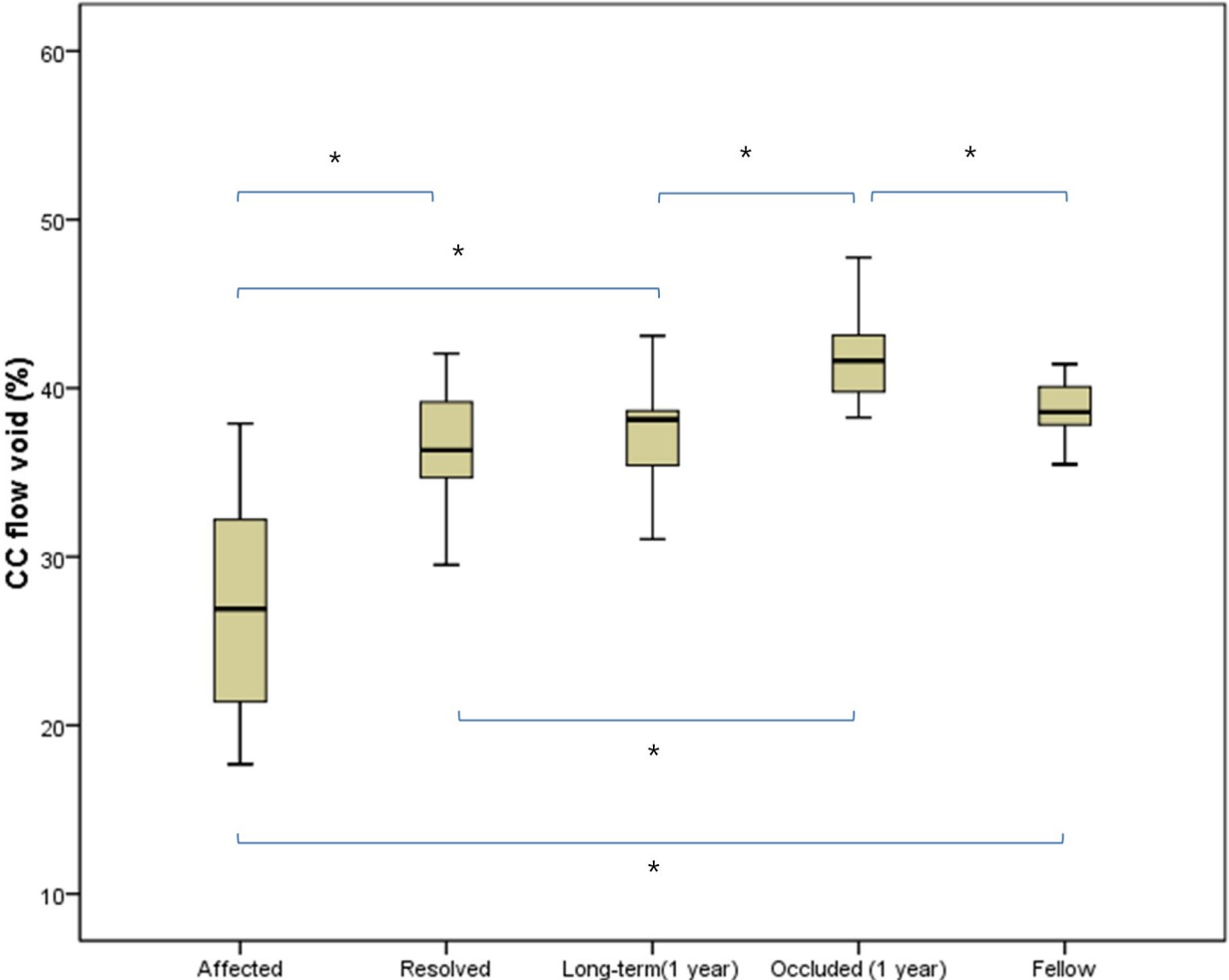

**Fig 2. Quadrant choriocapillaris (CC) flow voids (%) in the OCTA images as box-plots.** The values of each quadrant CC flow were presented with the following sequence: The uninvolved quadrant of acute BRVO-affected eyes, that of resolved eyes after 3-months, 1-year disease onset, the involved (occluded) quadrant of resolved eyes after 1-year disease onset, and the contralateral quadrant of fellow eye. The asterisk indicates significant *P*-value (<0.05). Paired t-test was used.

**Table 3. Univariate regression analysis results between choriocapillaris (CC) flow voids and several OCT parameters & the number of injections.**

| CC Flow voids | Opposite quadrants CC flow void | | | Occluded quadrants CC flow void | | |
|---|---|---|---|---|---|---|
| | (Acute affected) | | | (long-term resolved after 1 year) | | |
| | P value | $R^2$ | Standardized ß | P value | $R^2$ | Standardized ß |
| Initial CMT | **0.025** | 0.172 | -0.415 | 0.617 | 0.009 | 0.097 |
| Initial CT | 0.998 | 0.000 | -0.001 | 0.274 | 0.044 | -0.210 |
| Fellow CT | 0.783 | 0.003 | -0.053 | **0.029** | 0.164 | -0.405 |
| Fellow CVI | 0.355 | 0.032 | -0.178 | 0.194 | 0.062 | -0.248 |
| Total intravitreal injection | **0.031** | 0.160 | 0.400 | 0.961 | 0.000 | -0.010 |

ß, regression coefficient. p-values that were statistically significant are highlighted in bold. (p<0.05). CT: choroidal thickness; CVI: choroidal vascularity index.

Although the extracellular fluid inflow from retina to choroid, previously suggested by Aribas et al. [7], could generally aggravated the choroidal blood flow, it could also be speculated that our homeostatic system works to preserve the function of the outer retina by increasing the CC flow to a greater extent within the relatively intact neighboring area. Furthermore, the result that the CC flow void of the occluded quadrant after 1-year disease onset is higher than that of the fellow or any other acute, resolved uninvolved quadrant confirmed that the CC flow reduction was limited to the area where the actual vascular obstruction occurred. Borrelli et al. [18] hypothesized that if choroidal volume exceeded the specific threshold levels, fenestrated intravascular hydrostatic pressure would be increased in the CC, leading to CC hypoperfusion. The fact that CC hypoperfusion occurred only in a localized area is consistent with the results of Kim et al.'s report that CT significantly increased especially more in occlusive lesions of BRVO [5].

The regression analyses between several parameters and CC flow voids of acute uninvolved/long-term-resolved involved (occluded) quadrants showed some remarkable results. The p-value of acute uninvolved quadrant CC flow void with initial CMT was statistically significant (P = 0.025), suggesting that the more severe the ischemia or ME, the more active the compensation was in the surrounding area. More elaborate research should be done to elucidate whether the compensation was due to the highly concentrated VEGF from retina or the diminished sympathetic activity from the choroidal congestion, possibly caused by a rush of extracellular fluid. Second, the result that the long-term resolved BRVO involved (occluded) quadrant CC flow void decreased as the fellow eyes' CT increased could be demonstrated that the BRVO CC damage could be alleviated when an individual had a larger inherent choroidal volume capacity. An et al. [4] reported that BRVO affected/fellow eye CT ratio was associated with VEGF concentration and thicker CT might contribute the visual gain with the decrease in CMT after anti-VEGF injection. As for the mechanism suggested by Aribas et al. [7], when there is a fluid inflow from the retina due to retinal vascular obstruction, the choroidal flow congestion with delayed CC flow would occur, the degree of which would be determined according to the choroidal volume capacity [19]. Based on our previous report [6], we expected that fellow eye CVI would do a role for CC flow changes in the occluded areas, however, there was no significant p-value between CVI and any CC flow voids. The difference would come from the fact that the previous study selected the patient who did not show macula-involving hemorrhage in BRVO, and analyzed visual acuities of longer duration of illness. As we analyzed the only quadrant CC of 4.5 x 4.5 mm macula scan, subfoveal CT would better reflect the CC changes in the obstructed area than the CVI, which was calculated from total choroid of 12 x 9 mm scan. The effect of choroidal large veins dilatation, well representing as CVI, on CC flow change in BRVO might be clearly understood when backed up with a large and well-designed study. The correlation between the number of injections and the CC flow void in occluded quadrant also did not show a statistically significant p-value, suggesting that the additional decrease in CC flow due to repeated injections, which can be inferred from the study of Julien et al. [20], was not confirmed in this study. However, the meaningful p-value of comparison between the number of injections and the void in uninvolved quadrant at the acute phase could be interpreted that proper compensations of surrounding area could affect BRVO prognosis, or recurrence rates, especially in terms of treatment intensities.

There were some limitations to this study. CC flow voids of obstructed quadrant could not be continuously measured due to hemorrhage masking effect at the acute phase. CC flow changes from the longer term perspective could not be confirmed with only 1-year of observation. The eyes of BRVO that occurred at the peripheral side, not accompanying macular hemorrhage or edema, which accounts for a significant portion in the real world, were excluded

from the study. The lower lateral resolution of Topcon SS-OCTA than PLEX Elite 9000, and a small number of subjects were other possible shortcomings in the current study.

In conclusion, we identified the pattern of CC flow changes in the surrounding macular area of BRVO by analyzing serial quadrant OCTA macula scans in this study. We believe that we were able to get closer to the changes of choroidal flow in eyes with BRVO with our novel results.

## Supporting information

**S1 File. Dataset.**
(XLSX)

## Author Contributions

**Conceptualization:** Bo-Een Hwang, Young-Hoon Park.

**Data curation:** Bo-Een Hwang, Joo-Young Kim, Mirinae Kim, Young-Geun Park, Young-Hoon Park.

**Formal analysis:** Bo-Een Hwang, Jae-Hyuck Kwak, Joo-Young Kim, Rae-Young Kim, Mirinae Kim, Young-Geun Park, Young-Hoon Park.

**Funding acquisition:** Bo-Een Hwang, Young-Hoon Park.

**Investigation:** Bo-Een Hwang, Jae-Hyuck Kwak, Joo-Young Kim, Rae-Young Kim, Mirinae Kim, Young-Geun Park, Young-Hoon Park.

**Methodology:** Bo-Een Hwang, Jae-Hyuck Kwak, Joo-Young Kim, Rae-Young Kim, Mirinae Kim, Young-Geun Park, Young-Hoon Park.

**Project administration:** Bo-Een Hwang, Joo-Young Kim, Mirinae Kim, Young-Geun Park, Young-Hoon Park.

**Resources:** Bo-Een Hwang, Joo-Young Kim, Rae-Young Kim, Mirinae Kim, Young-Geun Park, Young-Hoon Park.

**Software:** Bo-Een Hwang.

**Supervision:** Bo-Een Hwang, Jae-Hyuck Kwak, Joo-Young Kim, Rae-Young Kim, Mirinae Kim, Young-Geun Park, Young-Hoon Park.

**Validation:** Bo-Een Hwang, Joo-Young Kim, Mirinae Kim, Young-Geun Park, Young-Hoon Park.

**Visualization:** Bo-Een Hwang, Joo-Young Kim, Rae-Young Kim, Mirinae Kim, Young-Geun Park, Young-Hoon Park.

**Writing – original draft:** Bo-Een Hwang.

**Writing – review & editing:** Bo-Een Hwang, Jae-Hyuck Kwak, Mirinae Kim, Young-Hoon Park.

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
