## [Decision Letter · Decision Letter 0]

26 Oct 2022

PONE-D-22-16909Serial choriocapillaris flow changes in eyes with branched retinal vascular obstruction (BRVO).PLOS ONE

Dear Dr. Park,

Thank you for submitting your manuscript to PLOS ONE. After careful consideration, we feel that it has merit but does not fully meet PLOS ONE’s publication criteria as it currently stands. Therefore, we invite you to submit a revised version of the manuscript that addresses the points raised during the review process.

Please assess the variability between observers. Please reformat Tables 1 and 2. Pleasentighten up your Discussion so that its logic is clear.

We look forward to receiving your revised manuscript.

Kind regards,

Alfred S Lewin, Ph.D.

Section Editor

PLOS ONE

Journal Requirements:

“Funding resource: This study was supported by the Basic Science Research Program through the National Research Foundation of Korea (NRF-2020R1F1A1074898).”

“Competing interests: None of the authors have any conflicting interests to disclose.”

4. Please include a copy of Table 3 which you refer to in your text on page 10.

Reviewers' comments:

Reviewer's Responses to Questions

**Comments to the Author**

1. Is the manuscript technically sound, and do the data support the conclusions?

Reviewer #1: Yes

2. Has the statistical analysis been performed appropriately and rigorously? 

Reviewer #1: Yes

3. Have the authors made all data underlying the findings in their manuscript fully available?

Reviewer #1: Yes

4. Is the manuscript presented in an intelligible fashion and written in standard English?

Reviewer #1: Yes

5. Review Comments to the Author

Reviewer #1: Dr. Young-Hoon Park and his colleagues evaluated the CC flow changes in branched retinal vascular obstruction (BRVO) on OCTA. I think the study provided meaningful guidance for the changes of choroidal flow in eyes with BRVO. However, this paper has some minor mistakes which need to be modified.

1. In Methods about the statistical analysis, I think there should be the assessment of inter-oberserver reliability.

2. The structure of Table 1 and Table 2 should be modified according to the standards.

3. In Discussion, the structure needs some adjustments, for the logic is confusing to some extent.

6. PLOS authors have the option to publish the peer review history of their article (what does this mean?). If published, this will include your full peer review and any attached files.

Reviewer #1: No

---

## [Author Response · Author response to Decision Letter 0]

5 Nov 2022

Dear Editors,

Thank you for giving us the opportunity to submit a revised draft of our manuscript titled “Serial choriocapillaris flow changes in eyes with branched retinal vascular obstruction (BRVO)” to PLOS ONE. We appreciate the time and effort that you and the reviewers have dedicated to providing your valuable feedback on the manuscript. We are grateful to the reviewers for their insightful comments on the paper. We have been able to incorporate changes to reflect most of the suggestions provided by the reviewers. We have highlighted those changes within the manuscript with red color.

Here below are point-by-point responses to the reviewers’ comments and concerns.

Comments from Reviewer #1

The authors would like to thank the reviewer for taking the time to review our paper. We have made corrections and clarifications in the manuscript after going over the comments about limitations.

Comment 1: In Methods about the statistical analysis, I think there should be the assessment of inter-oberserver reliability.

Response: Thank you for your thoughtful comment. The intraclass correlation coefficient (ICC) analysis was performed to calculate intergrader reliability. ICC was 0.972 for single measures and 0.986 for average measurements (p < 0.001). We clarified these contents in the revised manuscript.

Comment 2: The structure of Table 1 and Table 2 should be modified according to the standards.

Response: Thank you for your thoughtful comments. We respected your opinion, and we made a modification of the table contents to better fit the standards.

Comment 3: In Discussion, the structure needs some adjustments, for the logic is confusing to some extent.

Response: We acknowledge that the explanation and logics we made seemed to cause confusion. We corrected the discussion contents to be more understandable. Thank you.

We look forward to hearing from you in due time regarding our submission and to responding to any further questions and comments you may have. 

Sincerely,

Young-Hoon Park, MD, PhD

Department of Ophthalmology and Visual Science, Seoul St. Mary's Hospital, College of Medicine, The Catholic University of Korea, 222 Banpo-daero, Seocho-gu, Seoul 06591, Republic of Korea

E-mail: parkyh@catholic.ac.kr

---

## [Editor Report · Decision Letter 1]

8 Nov 2022

Serial choriocapillaris flow changes in eyes with branched retinal vascular obstruction (BRVO).

PONE-D-22-16909R1

Dear Dr. Park,

We’re pleased to inform you that your manuscript has been judged scientifically suitable for publication and will be formally accepted for publication once it meets all outstanding technical requirements.

Kind regards,

Alfred S Lewin, Ph.D.

Section Editor

PLOS ONE
---

## [Editor Report · Acceptance letter]

10 Nov 2022

PONE-D-22-16909R1 

Serial choriocapillaris flow changes in eyes with branched retinal vascular obstruction (BRVO). 

Dear Dr. Park:

I'm pleased to inform you that your manuscript has been deemed suitable for publication in PLOS ONE. Congratulations! Your manuscript is now with our production department. 

Kind regards, 

on behalf of

Dr. Alfred S Lewin 

Section Editor

PLOS ONE